# The Identification of Nuclear FMRP Isoform Iso6 Partners

**DOI:** 10.3390/cells12242807

**Published:** 2023-12-09

**Authors:** Nassim Ledoux, Emeline I. J. Lelong, Alexandre Simard, Samer Hussein, Pauline Adjibade, Jean-Philippe Lambert, Rachid Mazroui

**Affiliations:** 1Centre de Recherche du CHU de Québec—Université Laval, Axe Oncologie, Département de Biologie Moléculaire, Biochimie Médicale et Pathologie, Faculté de Médecine, Université Laval, Québec, QC G1V 0A6, Canada; nassim.ledoux@crchudequebec.ulaval.ca (N.L.); emeline.lelong@crchudequebec.ulaval.ca (E.I.J.L.); alexandre.simard.15@ulaval.ca (A.S.); samer.hussein@crchudequebec.ulaval.ca (S.H.); pauline.adjibade.1@ulaval.ca (P.A.); 2Centre de Recherche du CHU de Québec—Université Laval, Axe Endocrinologie et néphrologie, Département de Médecine Moléculaire, Faculté de Médecine, Université Laval, Québec, QC G1V 0A6, Canada; jean-philippe.lambert@crchudequebec.ulaval.ca; 3PROTEO, Le Regroupement Québécois De Recherche Sur La Fonction, L’ingénierie et Les Applications des Protéines, Université Laval, Québec, QC G1V 0A6, Canada

**Keywords:** FMRP, RNA-binding proteins, GFP-Trap, mass spectrometry, proteasome

## Abstract

A deficiency of FMRP, a canonical RNA-binding protein, causes the development of Fragile X Syndrome (FXS), which is characterised by multiple phenotypes, including neurodevelopmental disorders, intellectual disability, and autism. Due to the alternative splicing of the encoding *FMR1* gene, multiple FMRP isoforms are produced consisting of full-length predominantly cytoplasmic (i.e., iso1) isoforms involved in translation and truncated nuclear (i.e., iso6) isoforms with orphan functions. However, we recently implicated nuclear FMRP isoforms in DNA damage response, showing that they negatively regulate the accumulation of anaphase DNA genomic instability bridges. This finding provided evidence that the cytoplasmic and nuclear functions of FMRP are uncoupled played by respective cytoplasmic and nuclear isoforms, potentially involving specific interactions. While interaction partners of cytoplasmic FMRP have been reported, the identity of nuclear FMRP isoform partners remains to be established. Using affinity purification coupled with mass spectrometry, we mapped the nuclear interactome of the FMRP isoform iso6 in U2OS. In doing so, we found FMRP nuclear interaction partners to be involved in RNA processing, pre-mRNA splicing, ribosome biogenesis, DNA replication and damage response, chromatin remodeling and chromosome segregation. By comparing interactions between nuclear iso6 and cytoplasmic iso1, we report a set of partners that bind specifically to the nuclear isoforms, mainly proteins involved in DNA-associated processes and proteasomal proteins, which is consistent with our finding that proteasome targets the nuclear FMRP iso6. The specific interactions with the nuclear isoform 6 are regulated by replication stress, while those with the cytoplasmic isoform 1 are largely insensitive to such stress, further supporting a specific role of nuclear isoforms in DNA damage response induced by replicative stress, potentially regulated by the proteasome.

## 1. Introduction

The inactivation of the X chromosome-linked *FMR1* gene causes the development of fragile-X syndrome (FXS), the most common cause of inherited mental retardation and possibly autism [1,2]. *FMR1* codes for FMRP, whose main function is in RNA metabolism. Multiple FMRP isoforms are produced due to extensive alternative splicing of the primary transcripts of the *FMR1* gene. These consist of full-length isoforms that are predominantly cytoplasmic (cFMRP) and truncated FMRP nuclear isoforms (nFMRP), with the main characterised ones being iso1/7 and iso6/12, respectively [3]. As shown in Appendix A, iso1 and iso7 are almost identical, with the only difference being the skipping of exon 12 in iso7 [4], while the only difference between iso6 and 12 is the skipping of exon 12 in iso12 [5]. All FMRP isoforms, including iso1/7 and iso6/12, share the common amino acid sequence from amino acid 1 to 374, which includes the Agenet/Tudor motifs at the N-terminal [6], the nuclear localization signal [7] and the RNA-binding KH domain [8]. Due to alternative splicing, nFMRP contains a unique C-terminal (C-t) domain that lacks the putative cytoplasmic retention domain (CRD) [3]. This iso6-specific C-t domain also lacks the RGG motif required for the cytoplasmic functions of FMRP [9,10], including its phase separation activity [11] but contains a putative nuclear localization signal [3,4], potentially required for the nuclear localisation of nFMRP. Whether this unique C-t domain nFMRP provides specific FMRP functions is currently unknown. The most established function of FMRP is in translation regulation, either directly acting on ribosomes or indirectly by transporting mRNAs [12,13]. This translational role of FMRP was established based mainly on data obtained using strategies that either target all FMRP isoforms or express cFMRP [13], precluding the analysis of the contribution of nFMRP in processes regulated by FMRP. 

More recent studies also reported the role of mammalian FMRP in DNA damage response. In the first study [14], the authors concluded that FMRP is required for stress-induced DNA damage in mouse embryonic fibroblasts and HeLa cells, while the subsequent studies using either fibroblasts derived from FXS patients [15] or FMRP-depleted U2OS [16], reported an opposite effect. Moreover, in the above studies [14,15,16], the observed effects were attributed to full-length (iso1) FMRP, which is mainly cytoplasmic, while the role of nuclear FMRP isoforms was not investigated in either study. The possibility that FMRP plays nuclear functions is also consistent with the finding that the N-terminal part of FMRP binds NUFIP (nuclear FMRP interacting protein), an interaction that is potentially mediated by nuclear FMRP iso12 [17]. We recently discovered a nuclear function of human FMRP played by its nuclear isoforms [18]. We found that nFMRP contributes to maintaining genome stability in both U2OS and HeLa cells by antagonising the accumulation of genomic instability structures, consisting of aberrant RNA-containing DNA bridges, preventing DNA damage and associated cell death [18]. Thus, in addition to its well-established role in regulating translation, FMRP also has nuclear functions, played specifically by its nuclear isoforms. Since nFMRP shares similar RNA-binding motifs with cFMRP, binds RNA, and associates with the nuclear foci called RNA Cajal bodies in various human somatic cells [3], it may play RNA-based nuclear functions involving specific interactions. In this study, by interrogating the U2OS proteome, we report the first nFMRP interactome. nFMRP partners have activities in both RNA and DNA nuclear processes, including transcription, splicing, RNA processing and modification, chromatin remodeling and DNA damage response. The finding that nFMRP, but not cFMRP, forms complexes with proteasomal proteins is intriguing, raising the possibility of a functional novel interplay between FMRP and nuclear proteasome. 

## 2. Materials and Methods

### 2.1. Cell Lines, Cell Culture, and Drug Treatment

Human osteosarcoma (U2OS) cell lines were propagated and maintained in DMEM medium supplemented with 5% fetal bovine serum and antibiotics (100 units/mL penicillin, 50 mg/mL streptomycin). Cells were maintained in a humidified incubator with 5% CO_2_ at 37 °C. Aphidicolin was obtained from Sigma-Aldrich (Oakville, ON, Canada) and stored at –20 °C. For treatments, cells were first washed and incubated in drug-free media for 2 h. Aphidicolin (0.3 μM) was then added to the media for the indicated periods. MG132 was obtained from Selleck Chemicals (Houston, TX, USA) and stored at –80 °C. For treatments, cells were first washed and incubated in drug-free media for 2 h. MG132 (10 μM) was then added to the media for the indicated periods. Cycloheximide was obtained from Sigma and stored at –20 °C. For treatments, cells were first washed and incubated in drug-free media for 2 h. Cycloheximide (50 μg/mL) was then added to the media for the indicated periods. Puromycine was obtained from Biobasic (Markham, ON, Canada) and stored at –20 °C. For puromycin treatment, the drug (50 μg/mL of puromycin) was added to the media for the last 5 min of the experiment.

### 2.2. Antibodies

Anti-α-tubulin, anti-GFP, anti-TOP2A and anti-Aurora B were obtained from Abcam. Anti-p53 was obtained from Santa Cruz Biotechnology (Dallas, TX, USA), and anti-PSMD and -PSMC were obtained from Proteintech (Fraunhoferstr, Germany). 

### 2.3. Confocal Microscopy

For microscopy, the cells were fixed with Acetone/Methanol or 4% paraformaldehyde and then permeabilized with PBS-containing 0.25% Triton X-100. For cell visualisation and data quantification, images were acquired through an LSM 900 laser scanning confocal microscope (Zeiss, Toranto, ON, Canada) equipped with Zen blue 3.5 software for image acquisition and processing. 

### 2.4. GST-Pull Down and GFP-Trap Assays

GST-pull down was performed essentially as described [19]. Briefly, purified GST-iso6 and the control GST were immobilised on glutathione agarose beads and incubated with a U2OS total cell extract prepared via lysis in a standard lysis buffer (50 mM Tris-HCl (pH 7.4); 150 mM NaCl; 0.5% NP-40 substitute; 100 mM PMSF, and protease inhibitors) for 1 h at 37 °C. The beads were washed in lysis buffer, boiled at 95 °C in 1x sodium dodecyl sulfate (SDS) loading buffer for 10 min, and separated on SDS/polyacrylamide gels. When indicated, the beads were incubated 15 min at room temperature with RNase A (10 μg/mL) before processing. 

For GFP-Trap, U2OS stably expressing GFP or GFP-iso6 was lysed in 1 mL lysis buffer (50 mM Tris-HCl pH 7.4, 150 mM NaCl, 1 mM MgCl2; 0.5% NP-40 substitute, 0.25 mM phenylmethylsulfonyl fluoride, 0.1 mM dithiothreitol, and a protease inhibitor cocktail added immediately prior to processing). To aid in lysis, the cells were resuspended 15–20 up and down using a syringe with a 27G needle and then incubated on ice for 15 min. After centrifugation, the supernatant of the lysate was incubated for 2 h at 4 °C with GFP-Trap-agarose beads (Proteintech). The beads were then washed three times in lysis buffer, boiled at 95 °C in Laemmli buffer for 10 min, and separated on SDS/polyacrylamide gels.

### 2.5. Stable GFP Transfection

mEGFP, mEGFP-iso6, and mEGFP-iso1 sequences were first ligated into the *Age*I and *PST*I restriction sites of a pLJM1 plasmid (Addgene). The pLJM1-mEGFP-iso6ΔC-t plasmid was obtained from NorClone (London, ON, Canada). To generate lentiviral-mEGFP, mEFFP-iso6, mEGFP-iso6ΔC-t, and mEGFP-iso1 particles, we transfected HEK 293T cells with the pLJM1-mEGFP, pLJM1-mEGFP-iso6, pLJM1-mEGFP-iso6ΔC-t, and pLJM1-mEGFP-iso1 plasmids, respectively, together with psPAX2 packaging and pMD2.G envelope plasmids (Addgene, Watertown, MA, USA). The medium was replaced, and twenty-four hours later, lentiviral particles were harvested, filtered through 0.45 μm filters, and supplemented with 8 μg/mL polybrene (Sigma). The collected viruses were used to transduce U2OS. U2OS stably expressing mEGFP, mEGFP-iso6, mEGFP-iso6ΔC-ter, and mEGFP-iso1 was then obtained via puromycin resistance selection.

### 2.6. Affinity Purification and MS Experiments

GFP AP-MS was essentially carried out as described [20]. Briefly, stable U2OS from either four or two 150 mm plates was lysed in 1.5 mL lysis buffer (50 mM HEPES-NaOH pH 8.0, 100 mM KCl, 2 mM ethylenediaminetetraacetic acid, 0.1% NP-40 substitute, and 10% glycerol, with 1 mM phenylmethylsulfonyl fluoride, 1 mM dithiothreitol, and Sigma-Aldrich protease inhibitor cocktail (1:500) added immediately prior to processing). To optimise lysis, the cells were frozen on dry ice, thawed in a 37 °C water bath, and returned to ice before gentle sonication using a Q125 sonicator (QSONICA) for three 10 secs pulse at an amplitude of 0.35. The lysates were incubated with turbonuclease (100 units, Sigma-Aldrich, T4332) for 1 h at 4 °C on a nutator and centrifuged at 20,817× *g* for 20 min at 4 °C, and the resulting supernatant was added to 25 μL of GFP-Trap magnetic agarose beads (ChromoTek, Fraunhoferstr, Germany; gtma-10) prewashed with lysis buffer. GFP pull down was performed at 4 °C for 2 h with rotation. The beads were pelleted via centrifugation (100× *g* for 1 min) and magnetized, and the unbound lysate was aspirated and kept for analysis. The beads were processed as previously described to elute the bound proteins, which were transferred to a fresh tube. Peptide samples were stored at −80 °C until MS analysis.

### 2.7. Data-Dependent Acquisition MS

MS analyses were performed at the Proteomics Platform of the Quebec Genomics Center. Peptide samples were separated via online reversed-phase nanoscale capillary liquid chromatography and analysed using electrospray MS/MS. The experiments were performed with a Dionex UltiMate 3000 RSLCnano chromatography system (Thermo Fisher Scientific, Waltham, USA) connected to an Orbitrap Fusion mass spectrometer (Thermo Fisher Scientific, Waltham, USA) equipped with a nanoelectrospray ion source. The peptides were trapped at 20 μL/min in a loading solvent (2% acetonitrile, 0.05% TFA) on an Acclaim 5μm PepMap 300 μ-Precolumns Cartridge Column (Thermo Fisher Scientific) for 5 min. Then, the precolumn was switched online with a laboratory-made 50 cm × 75 μm internal diameter separation column packed with ReproSil-Pur C_18_-AQ 3-μm resin (Dr. Maisch HPLC), and the peptides were eluted with a linear gradient of 5–40% solvent B (A: 0.1% formic acid, B: 80% acetonitrile, 0.1% formic acid) over 90 min at 300 nL/min. Mass spectra were acquired in the data-dependent acquisition mode using Thermo XCalibur software version 3.0.63. Full-scan mass spectra (350–1800 *m/z*) were acquired in the Orbitrap using an AGC target of 4 × 10^5^, a maximum injection time of 50 ms, and a resolution of 120,000. Internal calibration using lock mass on the *m/z* 445.12003 siloxane ion was used. Each MS scan was followed by the acquisition of the fragmentation spectra of the most intense ions for a total cycle time of 3 s (top speed mode). The selected ions were isolated using the quadrupole analyser in a window of 1.6 *m/z* and fragmented through the use of higher-energy collision-induced dissociation at 35% collision energy. The resulting fragments were detected through the application of the linear ion trap at a rapid scan rate with an AGC target of 1 × 10^4^ and a maximum injection time of 50 ms. The dynamic exclusion of previously fragmented peptides was set for a period of 20 s and a tolerance of 10 ppm.

### 2.8. Protein Identification

MS data were stored, searched, and analysed using the ProHits laboratory information management system [21]. Thermo Fisher Scientific RAW mass spectrometry files were converted to mzML and mzXML using ProteoWizard (3.0.4468; [22]). The mzML and mzXML files were then searched using Mascot (v2.3.02) and Comet (v2012.02 rev.0) against the RefSeq database (version 57, 30 January 2013) acquired from NCBI, containing 72,482 human and adenovirus sequences supplemented with “common contaminants” from the Max Planck Institute (http://lotus1.gwdg.de/mpg/mmbc/maxquant_input.nsf/7994124a4298328fc125748d0048fee2/$FILE/contaminants.fasta; accessed on 19 July 2022), and the Global Proteome Machine (GPM; http://www.thegpm.org/crap/index.html; accessed on 19 July 2022). Charges of +2, +3, and +4 were allowed, and the parent mass tolerance was set at 12 ppm, while the fragment bin tolerance was set at 0.6 amu. Deamidated asparagine and glutamine and oxidized methionine were allowed as variable modifications. The results from each search engine were analysed through the Trans-Proteomic Pipeline (v4.6 OCCUPY rev 3) [23] via the iProphet pipeline [24]. To identify significant interaction partners, we used SAINTexpress ([25]; version 3.6.1) using default parameters. The results of these analyses can be found in Appendix A.

### 2.9. Experimental Design and Statistical Rationale for MS Experiments

For each analysis, at least two biological replicates of each bait were processed independently, with negative controls included in each batch of processed samples. The order of sample acquisition on the LC-MS/MS system was randomized. Statistical scoring was performed against the negative controls using Significance Analysis of INTeractome ((Teo et al., 2014 [25]); SAINTexpress 3.6.1), as defined in Appendix A. The average SAINTexpress score was used to determine the Bayesian false discovery rate (FDR), which requires a high confidence interaction to be detected in both biological replicates to pass our 1% FDR significance threshold.

**MS Data Visualisation:** We used ProHits-viz [26] to generate scatter and dot plots. To enhance the illustrations, individual nodes or dots were manually arranged in some figures. All MS files used in this study were deposited to MassIVE (http://massive.ucsd.edu) and can be accessed at ftp://MSV000092986@massive.ucsd.edu access on 4 December 2023. The password to access the MS files prior to publication is “FMRP”. The usernames for web access prior to publication are “MSV000092986_reviewer”. Additional details (including MassIVE accession numbers and FTP download links) can be found in Appendix A.

**MS Functional annotation and correlation:** The lists of differentially associated proteins were analysed using the functional annotation tool of the Database for Annotation Visualization and Integrated Discovery (DAVID; v 2021). Gene ontology terms and biological functions with a *p* value < 0.05, FDR < 0.01, ease at 0.01 and with at least 10 genes per go-terms were considered significantly over-represented within a gene list. Lists can be found in Appendix A.

We used Excel (Microsoft Office 360 Professionnel plus 2013) to generate Scatter plots. Concordances between different gene lists were assessed via Spearman correlations.

## 3. Results

### 3.1. Identification of nFMRP Partners

Because nFMRP is mainly in the nucleus, involved with nuclear DNA damage responses, and contains a unique C-t region not found in cFMRP, we reasoned that it may bind unique and nuclear partners required for its localisation and function. However, the identity of the nFMRP-interactome remained unknown. To identify such partners, we used an affinity purification approach coupled with mass spectrometry, which allowed for an unbiased analysis of nFMRP interacting proteins. To do so, we used U2OS stably expressing iso6 fused to GFP, and as controls we used GFP-free U2OS and U2OS-expressing iso1, as we have previously described [18]. As shown in Figure 1A, our Western blot analyses using anti-GFP antibodies show that GFP-iso6 is moderately expressed compared to GFP-iso1, probably reflecting the low expression of endogenous nFMRP and the difficulties in detecting its expression via a Western blot analyses of total extracts prepared from either HeLa or U2OS [3,18]. Due to the lack of suitable antibodies that can detect quantitatively and specifically endogenous iso6 in Western blot analyses, we could not provide comparison between GFP-FMRP iso6 and endogenous FMRP iso6 levels to demonstrate that GFP-FMRP expression is within a physiological range. However, our immunofluorescence experiments (images are provided to show differential localisation and not differential expression between iso6 and iso1) confirmed the nuclear and cytoplasmic localisation of GFP-iso6 and iso1 (Figure 1B), respectively. We have previously shown that the overexpression of FMRP induces FMRP-containing RNA granules, such as stress granules [27,28], which may affect the activity and localisation of the protein. To avoid the induction of this phenomenon due to the excessive expression of GFP proteins, we have selected for our studies U2OS clones that do not show such a formation of FMRP granules (Figure 1B), indicating moderate expression. Previous studies have reported the association of GFP-iso6 with Cajal bodies formed in either HeLa or mouse embryonic fibroblasts [3]. This association was observed upon acute (6 h postransfection) expression of the protein, while the presence of the protein in Cajal bodies upon prolonged expression was not analysed. As expected, we constantly observed the association of a transiently (6 h postransfection) expressed GFP-iso6 in U2OS with Cajal bodies. Using our U2OS stably expressing GFP-iso6, we occasionally observed GFP-iso6 localised in Cajal bodies, though this association with Cajal bodies was not consistently evident, probably reflecting either the possibility that the expression level of iso6 may affect that association or that the association of the protein with those foci in U2OS is transiently modulated by specific interactions. 

We thus first performed pull-down on whole cell extracts of U2OS-expressing GFP-iso6 using the GFP-nanoTrap and quantified the bound proteins using mass spectrometry. To identify the protein significantly enriched in our nFMRP purifications over our controls, we employed the SAINTexpress algorithm [25] and enforced a 1% false discovery rate (FDR); see methods for all details. Proteins that are similarly identified in the GFP-free U2OS control pull-down were eliminated, and those found exclusively in the two GFP-iso6 pull-down replicates were further filtered. The specificity of each identified interaction partner against our control was also evaluated by examining its fold change by determining the ratio of the abundance in the GFP-iso6 pull-down relative to the controls and by the average spectral count. This first set of data revealed that among the 1500 identified interactions, about 512 occur with less than 1% FDR and have a fold change greater than 1.5, potentially representing both transient and stable nFMRP-high confident partners (Appendix A and Figure 1C). As expected, top-hits include FXR1/2P (Appendix A and Table 1), the two well-characterised partners that interact with FMRP through its common N-terminal region [29], potentially corresponding to the respective nuclear isoforms [30,31]. Gene ontology annotations and analysis show that the majority of the 512 co-purified proteins with GFP-iso6 were nuclear regulators of specific biological RNA and DNA processes (Table 1 and Appendix A), including RNA splicing and processing (i.e., SRSF, PRP, CPSF1), RNA modification and RNA structure (i.e., YTHDF2, DDX, TRMTL1), ribosome biogenesis and maturation (i.e., WDR, MRT, NOP), RNA synthesis (i.e., POLR, CEBPZ, SUPT6H), DNA replication and DNA repair (i.e., BRCA2, TOP2A/B, SMARCA), and chromatin remodeling and chromosome condensation and segregation (i.e., H2A, AURKB, RCC1). Our data also revealed regulators and components of the ubiquitin–proteasome system (i.e., PSMDs, PSMCs, and UBAP2L), which is responsible for protein quality control processes, among the top binders of GFP-iso6. While this finding is intriguing, it is consistent with a recent study showing that FXR1P, the FMRP homolog, co-precipitates with the proteasome in neuroblastoma cells using ubiquitin-like domain beads [32]. As of now, it remains unknown whether this co-precipitation reflects interactions between FXR1P with proteasome components and, importantly, whether FMRP binds proteasome components. Our data show that while most PSMD and PSMC (PSMD 1–14 and PSMC 1–6) of the 19S regulatory particle of the 26S proteasome significantly co-purified with GFP-iso6, none of the 20S catalytic particles of the proteasome were detected in that pull-down complex, suggesting a possible specific function of nFMRP in regulating the proteasome through interaction with the 19S particle. 

To validate our mass spectrometry data, we first used GFP-Trap in the case of affinity-purified GFP-iso6 interactors, followed by Western blot analysis, confirming interactions of GFP-iso6 with the tested interactors, including PSMD2 and PSMD6 (Appendix A). To further confirm the GFP-Trap data, we employed in vitro GST pull-down assays assessing the interaction between purified GST-iso6 and selected hits, including AURBK, TOP2A, PSMD2, and PSMD6, which have not previously been described as FMRP partners. For these experiments, U2OS extracts were incubated with GST-iso6 immobilised on glutathione beads, and bound proteins were eluted and analysed through the use of a Western blot with specific antibodies. As shown in Figure 1D, the tested proteins were readily recovered with GST-iso6 but not with the control GST, albeit with different efficiencies, validating our GFP-Trap data. RNAse treatment of U2OS extracts does not affect GST-iso6 interactions with specific partners, such as PSMD2 and PSMD6 (Appendix A), suggesting that nFMRP interaction, with at least a subset of partners, are RNA independent. Collectively, through this first set of experiments, we revealed nFMRP interactions involved in RNA, DNA, and protein processes.

### 3.2. Identification of Potential nFMRP-Specific Partners

As mentioned above, nuclear and cytoplasmic FMRP isoforms share a common N-terminal region (Appendix A) that has been shown to mediate FMRP interactions and thus may bind similar partners. Thus, we sought to examine interactions with nuclear and cytoplasmic FMRP isoforms using the similar GFP-Trap strategy as above. In this case, we performed pull-downs on whole-cell extracts prepared from a similar number of U2OS cells expressing either GFP-iso6 or -iso1, quantifying GFP-nanoTrap-bound proteins to either iso6 or iso1 by LC-MS/MS. However, it should be noted that, as mentioned above (Figure 1A), GFP-iso6 is much less abundant compared to iso1, thus precipitating fewer proteins in similar conditions. In this experiment, assessing differential interactions between iso6 and iso1, we employed half the number of cells used in the experiments for GFP-iso6 pull-down, as described in Figure 1C, while including six independent sets of GFP-free U2OS controls compared to two sets used in the GFP-Trap reported in Figure 1. This experimental design remained effective at eliminating weak and potentially transient interactions in these stringent conditions. These data revealed that under our stringent conditions, ~700 proteins were pulled down with iso1 (Appendix A and Figure 2A). Among these, 77 are ≤1% FDR and have a fold change superior to 3, potentially representing proteins that stably associate with the cytoplasmic FMRP isoforms (Appendix A and Figure 2A). As expected, and similar to the data obtained with GFP-iso6 (Appendix A and Figure 1C), the two top hits identified in our GFP-iso1 screen are FXR1/2P (Appendix A), which are known to function as a complex with FMRP, supporting the validity of our proteomic data. Additional hits include RNA-binding proteins involved in translation regulation (i.e., PABC1, LARP1), in RNA modification (i.e., TRMT1L, DIMT1), which are components of RNA granules (i.e., G3BP1, STAU1) or act in RNA transport (i.e., CHTOP) (Appendix A). Due to the high stringency conditions used in our pull-downs, only 17 GFP-iso6 interaction partners were quantified with a ≤1% FDR among a total of 625 interactions identified (Appendix A and Figure 2B). These 17 proteins may represent the most prominent interaction partners of GFP-iso6. Consistently, and given the high stringency of the used parameters, both fold change and averaged spectral counts of scored GFP-iso6 interactions, including FXR1/2P, are significantly lower compared to those (Appendix A) scored in our first screens, precluding accurate comparison between the two sets of GFP-iso6 interactions. Thus, while mild stringency pull-down conditions revealed both modest (i.e., potentially transient) and strong (i.e., potentially stable) interactions with GFP-iso6 (Figure 1 and Appendix A), our high stringency conditions revealed FXR1/2P and proteasomal proteins, namely PSMD1, PSMD2, PSMD3, PSMD6, PSMD8, PSMD11, PSMD13, PSMD14, and PSMC4, as potentially the most stably associating partners with nFMRP (Table 2 and Appendix A). Interestingly, except for FXR1/2P, the majority of proteins that are scored as strong GFP-iso6 interactors (i.e., PSMDs, PSMCs) are not found in the GFP-iso1 pull-down (Appendix A), indicating specificity in our datasets. Similarly, proteins scored as strong GFP-iso1 interactors are either missing in the corresponding GFP-iso6 pull-down or have significantly higher fold change than controls compared to those found in the GFP-iso6 pull-down (Appendix A). Accordingly, a comparison between the full set of proteins precipitating with iso6 (625) and iso1 (705) shows an imperfect correlation in terms of interactions (Figure 2C), supporting differential interactions. Collectively, these data indicate that nFMRP can be engaged in specific interactions, potentially reflecting specific functions.

### 3.3. Regulation of nFMRP Interactions by Replicative Stress

Aphidicolin (APH) is an inhibitor of DNA polymerase that causes the formation of aberrant structures in replicating DNA, a phenomenon called replication stress, inducing the accumulation of DNA bridges and DNA damage [33,34,35]. We recently used APH as a source of DNA damage, showing that nFMRP efficiently antagonises their accumulation in U2OS [18]. This finding raises the possibility that replicative stress induced by APH treatment may dynamically modulate nFMRP interactions toward genome stability. To test if FMRP interactions are sensitive to replicative stress, U2OS expressing either iso6 or iso1, were treated with APH, and their whole extracts were used in the GFP-Trap tested for interactions under stringent conditions. Our data revealed that ~700 proteins in extracts prepared from U2OS-expressing GFP iso6 and treated with APH were pulled down with iso6 compared to GFP-free controls (Appendix A and Figure 3A). Among these, 43 have ≤1% FDR with a fold change above 10, reflecting quantitative interactions (Appendix A and Figure 3A). Based on fold changes, we did not observe an apparent effect of APH treatment on iso6 interactions with either FXR1P or FXR2P (Table 3 and Appendix A), while this APH treatment globally enhances iso6 interactions with the newly identified partners (Table 3, Figure 3B and Appendix A). Consistently, we found additional iso6 interactions with ≤1% FDR occurring in APH-treated U2OS as compared to mock-treated U2OS (Table 3), further indicating that APH treatment enhances iso6-specific interactions, rendering them sufficiently strong to be detected under stringent conditions. Among these enhanced interactions, AURBK [36], TOP2A [37], HP1BP3 [38], KIF22 [39], and ECT2 [40], are master factors involved in regulating genome stability. PAK1IP1 is a nucleolar protein dedicated to ribosomal RNA processing [41], while PCF11 and CPSF1 are canonical factors required for mRNA maturation [42]. AURBK is also among the nuclear kinases that affect the activity of partners through phosphorylation [43]. How APH treatment promotes those interactions remains unknown and will be explored in future investigations. Contrary to iso6 interactions, APH treatment slightly reduces the number of identified iso1 interactions, from 705 (Figure 2A) to 684 (Figure 3C), including those scored with ≤1% FDR, from 77 (Figure 2A) to 50 (Figure 3C). However, the majority (90%) of iso1 interactions found in mock-treated U2OS are affected neither positively nor negatively by APH treatment (Figure 3D and Appendix A), suggesting that replicative stress regulates FMRP interactions that are mainly mediated by its nuclear isoforms. As expected, with few exceptions, iso6 and iso1 interactions that occur in APH-treated cells are largely different (Appendix A).

### 3.4. Iso6 Binds Proteasomal Proteins, While Its Expression Is Regulated by the Proteasome

Our data described above identified most PSMD and PSMC components of the 19S regulatory complex of the proteasome precipitating with iso6 (Appendix A; Figure 1D), while no proteasomal components have been detected precipitating with iso1. These results raised the possibility that iso6, but not iso1, constitutes a proteasomal target limiting its expression, which may explain our data constantly showing that iso6 is less abundant than iso1 (Figure 1A and Figure 4A,B). To test this possibility, we treated U2OS-expressing GFP-iso6 and, as a control, those expressing GFP-iso1 with MG132, which blocks the activity of the proteasome [44], and the expression of the proteins were analysed using Western blot. As shown in Figure 4A, treatment with MG132 that inhibits the activity of the proteasome, which was confirmed by the enhanced expression of p53, had a minor (less than 1.5-fold) non-significative effect on the expression of GFP-iso1 (left panels), while it drastically increases (~8 fold) the expression of GFP-iso6 (right panels). The only difference between iso6 and iso1 is the presence of a C-t domain in iso6 that is different in iso1 [3]. We thus reasoned that this C-t domain may be responsible for iso6 targeting by the proteasome. To test this possibility, we generated U2OS-expressing GFP-iso6 lacking C-t (GFP-ΔC) and assessed its expression as above. As expected, we found that GFP-ΔC is significantly more expressed than GFP-iso6 (Figure 4B) and has both nuclear and cytoplasmic localisation (Figure 4C). These results support the role of the C-t domain in retaining the protein in the nucleus and its potential in terms of targeting the proteasome for degradation. Furthermore, the expression GFP-ΔC is not affected by MG132 treatment (Figure 4D), as is the case for GFP-iso6 (Figure 4A), indicating that the protein is refractory to proteasome targeting. We further assessed the degradation rate of both WT GFP-iso6 and GFP-ΔC by treating U2OS expressing either protein with cycloheximide that blocks the general translation. As shown in Figure 4E, the level of GFP-iso6 drops rapidly when general translation is inhibited (Figure 4F) upon treatment with cycloheximide, while a similar treatment had a marginal non-significative effect on the level of GFP-ΔC. Together, these results suggest that nFMRP is a labile protein that is targeted by the proteasome.

## 4. Discussion

Here, we report the interactome of nuclear FMRP isoforms. Iso6 partners include nuclear proteins involved in RNA and DNA processes, potentially reflecting various functions of the proteins in RNA synthesis and modification, DNA replication, and genome stability. Our finding that iso6 binds proteasomal proteins supports the possibility that the protein is a new proteasomal substrate, while it may also indicate a novel role of the protein in regulating the activity of a nuclear proteasome.

Our finding that iso6 pulled down with nuclear RNA-binding proteins, such as SRSF, CPSF1, YTHDF2, DDX, TRMTL1, WDR, POLR, CEBPZ, and SUPT6H (Appendix A and Table 1), is consistent with the possibility that the protein plays nuclear RNA functions, including splicing (i.e., via SRSF1 interaction), processing (i.e., via CPSF1 interaction), RNA modification (i.e., through YTHDF2 interaction), and synthesis (i.e., via CEBPZ binding), though experimental evidence supporting this assumption remains to be demonstrated. Nevertheless, the finding that nFMRP associates with the nuclear foci called RNA Cajal bodies in various human somatic cells [3,18] is also supportive of a functional role of the identified FMRP interaction with Cajal bodies-associated proteins, including SRSF, PRP, USP, and NOP [45,46], in RNA-based Cajal bodies functions, such as splicing and RNA processing. Thus, while nFMRP may regulate RNA expression, localisation, modification, and function through direct interaction via its KH domain, association with RNA-binding protein partners may be required when selecting RNA targets, stabilising interactions with RNA, and forming functional RNA complexes regulating the expression of target genes. However, at this stage, no RNA or gene targets of nFMRP have been identified. RNA omics approaches, such as RNA-CLIP [47] and RIP-seq [48], in nFMRP-expressing and -deficient cells should allow the identification of target RNAs whose activity and function are regulated by the protein and target genes whose encoded RNA abundance involves its activity and interactions. This is particularly important in terms of understanding the nuclear function of FMRP in neurons and how its deficiency may contribute to the development of FXS. However, to date, no study has been reported investigating either the expression, localisation, interactions, and functions of nFMRP in neurons and, thus, the role of such nuclear isoforms and their interactions with partners in neuron development and differentiation, whose alteration contribute to FXR, remains completely unknown. Among the cell models recently developed that have proven to be invaluable tools in investigating the molecular mechanisms underlying the role of cFMRP in neurone development and differentiation are human pluripotent stem cells (hPSCs). WT hPSC and *FMR1*-KO hPSCs lacking all FMRP isoforms and their neuron progenitors have been used in transcriptomic studies identifying differentially expressed genes encoding functions related to cell cycle regulation, chromatin assembly, oxidative stress, ATP production, ribosomal functions, neuronal differentiation, and synaptic functions [49]. Because *FMR1*-KO hPSCs lack all FMRP isoforms, it was not possible to address from these experiments the role of nFMRP in identified differential gene expression and neuron development. Using WT hPSCs lacking nFMRP and *FMR1*-KO hPSCs reprogramed with nFMRP, future studies combining the expression and localisation studies of nFMRP with transcriptomics, CLIP assays, and proteomics should allow us to define the contribution of nFMRP in neuronal development and differentiation through RNA-based nuclear functions.

An additional class of partners identified in our screen corresponds to proteins with activities in DNA replication, such as BRD4 [50], and DNA damage response, such as BRCA2 [51], chromatin remodeling, and segregation, i.e., AURKB [52] and anaphase genomic instability bridges, such as TOP2A [53] (Figure 1D, Table 1 and Table 3, Appendix A). These interactions also validated via the use of GST pull-down (Figure 1D), seem however transient under normal growth conditions as they are not readily detected under stringent conditions (Appendix A). However, these interactions appear to occur stably in APH-treated U2OS, suggesting that replicative stress promotes interactions of nFMRP with proteins involved in DNA damage response (Appendix A and Table 3). The finding that nFMRP associates with DNA bridges and chromatin segregation regulators, such as TOP2A and AURKB, is consistent with our recent study, implicating nFMRP in DNA damage response [18], associating with anaphase DNA bridges and downregulating their accumulation though the molecular mechanisms underlying the role of nFMRP in DNA bridge regulation remains elusive. Future experiments using depletion and mutagenesis studies should determine if TOP2A and AURKB are functional partners of nFMRP required for nFMRP-mediated downregulation of DNA bridges and associated DNA damage accumulation. Overall, our data that identified partners, such as TOP2A, AURKB, and BRCA2, provide potential avenues to exploring the role of these interactions in the control of genome stability through functional characterisation studies. Using hPSC-derived neuronal models, such investigations of the functional role of nFMRP-partner in genome stability may further allow a better understanding of the pathophysiology of FXS. 

While our study revealed nuclear interactions with iso6 occurring in U2OS, in vitro proteomic screenings were previously conducted by using GST pull-down with isolated N-terminal and C-terminal regions of full-length FMRP corresponding to the cytoplasmic isoforms as baits and total cell extracts from HeLa cells as the source of preys [54], listing a set of 102 interactions, with the majority corresponding to RNA-binding proteins involved in translation, RNA modification, RNA splicing, and processing. Among these interactions, ~5% are found in our list of 512 iso6 reported interactions (Appendix A), potentially corresponding to common partners. Previous quantification data showed that although full-length FMRP (i.e., iso1) is mainly cytoplasmic, a minor (~2–4%) fraction of this protein was inferred to be either nuclear [55] or associated with nuclear microtubules [3], which may thus engage in nuclear interactions and functions. In an attempt to identify nuclear proteins interacting with that minor nuclear fraction of full-length FMRP, Kieffer et al. [56] used in vitro GST pull-down with the N-terminal common part of FMRP on a nuclear fraction isolated from rat forebrains, coupled to quantitative mass spectrometry analysis. Using an adapted statistical workflow, they identified a total of 55 potential nuclear interactions they classified as either high- (proteins with a fold-change enrichment cutoff of >3.00) or medium- (proteins with a fold-change cutoff of >2.00) confidence interactions, with a majority corresponding to RNA-binding proteins involved in RNA splicing and processing, transcription, and mRNA transport. In this study, 10 (i.e., PPP1R10, UTP14a, POLDI3, ELAVL1, CHD4) out of these 55 rat FMRP interactions (Appendix A) were found in our list of 512 iso6 (Appendix A) reported interactions, potentially corresponding to conserved common partners. 

However, it is worth noting that several proteins scored as having significant interactions in the in vitro rat [56] and human pull-downs [54] were also found in our extended list of interactions, which we scored as low-confidence interactions as they did not pass our 1% FDR. These discrepancies in confidence may be due to the nature of assays used, i.e., in vitro GST-pull down using isolated parts of FMRP versus our in vivo GFP-Trap using WT iso6. However, we are aware that increasing the level of iso6 in U2OS by expressing GFP-iso6 may lead to an overestimation of interactions. Comparative studies assessing the expression of iso6 in U2OS-expressing GFP-iso6 relative to U2OS through the use of quantitative LC-MS/MS on isolated nuclei or via translatomic studies quantifying the association of iso6 RNA with translating polyribosomes may help address this issue; however, our immunofluorescence experiments (Figure 1B) suggest the moderate expression of GFP-iso6 in our U2OS clones we selected for interaction studies. Discrepancies in interaction confidences between the data obtained with in vitro and in vivo experiments may also be due to posttranscriptional modifications of FMRP negatively affecting in vivo interactions. Because a minor (2–4%) fraction of full-length FMRP (i.e., iso1) has been found to be nuclear [55], and a minimal sub-fraction of GFP-iso6 can also be detected in the cytoplasm (Figure 1B), both isoform types may be engaged in common but limited interactions in both the nuclei and the cytoplasm. Nevertheless, while nuclear and cytoplasmic FMRP isoforms share few common interactions, the majority seems to be specific to each type of isoform type.

To further assess this possibility, we performed stringent GFP-trap proteomics using either iso6 or iso1 as baits expressed in U2OS, scoring interactions occurring with high confidence. Our analyses (Figure 2B, Table 2, and Appendix A) indicate that iso6 stably associates with a limited set of proteins (see below), probably corresponding to transient and extensively regulated interactions. We also assessed iso1 interactions under similar stringent conditions (Appendix A, Figure 2A), revealing a list of 77 stable interactions, including well-established RNA-binding protein interactors, such as FXR1P, G3BP1, and PABPC1. Among these interactions, 20% (i.e., FXR1P, CIRBP, PRRC2C, MOV10, LARP1, PABPC, G3BP) were reported to occur in the previous in vitro affinity pull-down of total extracts prepared from HeLa cells using the N-terminal or the C-terminal region of the cytoplasmic FMRP isoforms as baits [54], indicating that a significant set of interactions with the cytoplasmic FMRP isoforms are conserved between different cell lines and conditions. However, a comparison between the full set of proteins precipitating with iso6 (625) and iso1 (704) shows a significant, though imperfect, correlation in terms of interactions (Figure 2C), indicating that iso6 can be engaged in specific interactions, potentially reflecting unique functions.

As mentioned above, we recently implicated nFMRP in DNA damage response, associating with anaphase DNA bridges, and downregulating their accumulation [18]. While the molecular mechanisms underlying the role of nFMRP in DNA bridges regulation remain elusive, our data identifying partners, including TOP2A, BRCA2, BRD4 and AURKB, with activities in DNA replication, DNA damage response, chromatin remodeling, and segregation, provide potential avenues to explore defining the role of these interactions in the control of genome stability through functional characterisation studies including the regulation of DNA bridges. Consistently, the finding that iso6 interactions, i.e., AURKB, TOP2A, and BRD4, are significantly promoted by APH treatment (Figure 3A,B, Table 3, and Appendix A) also supports the possibility of a functional role of these nFMRP interactions in dampening the effects of replicative stress, though direct evidence is still lacking. Nevertheless, these results indicate that nFMRP interactions are sensitive to stress-inducing DNA damage, further supporting the implication of nFMRP in DNA damage response. 

Among proteins that are specifically pulled down with iso6 (Appendix A) but not with iso1 (Appendix A) are PSMDs and PSMCs, the components of the 19S regulatory proteasome particle (RP), possibly reflecting interactions with a nuclear proteasome. Interactions of nFMRP with 19S components seem to be specific, as no PSMA component of the 20S catalytic particle of the proteasome was found to precipitate with iso6. Moreover, the finding that the majority of nFMRP interactions with 19S RP are maintained and even promoted during APH treatment (Table 3 and Appendix A) supports the possibility that such interactions may regulate nFMRP function in genome stability, which is consistent with the role of the nuclear proteasome in the quality control of the genome by regulating the stability of target proteins [57]. Alternatively, nFMRP may regulate the activity of the proteasome in the nucleus, affecting its function in either RNA or DNA processes. This is consistent with a recent study reporting that the depletion of FMRP using siRNAs that target all its isoforms increases in vitro proteasome activity [32], though it remains unknown if this effect is due to the loss of nFMRP, cFMRP or both. However, in this study [32], the authors excluded the possibility of an interaction between FMRP and the proteasome collected using the ubiquitin beads based on Western blot analyses of co-precipitating proteins. However, these Western blot analyses used antibodies designed to detect full-length FMRP (i.e., iso1), which is largely cytoplasmic. As far as we know, these antibodies do not detect nFMRP quantitatively in Western blot, which requires specific antibodies to be detected [3]. Nevertheless, the finding that full-length cytoplasmic FMRP is absent from proteasome precipitates is consistent with our data, excluding the interaction between iso1 and proteasomal components, supporting our conclusion that FMRP–proteasome interactions are mainly mediated by nFMRP. While we do not know if nFMRP regulates the proteasome activity, or vice versa, our data (Figure 4A,B) show that iso6, but not iso1, is significantly stabilised by MG132, strongly suggesting that nFMRP is a novel substrate of the proteasome, regulating its stability, providing a plausible explanation of the observed lower expression of nuclear FMRP isoforms as compared to those that are predominantly cytoplasmic. Consistently, our cycloheximide experiments show that acute inhibition of general translation results in a rapid loss of iso6 expression, potentially reflecting its rapid decay (Figure 4E). Deleting the C-t domain of iso6 results in a partial localisation of the protein in the cytoplasm (Figure 4C) and a partial rescue of its expression (Figure 4B) due to its stabilisation (Figure 4D,E). These studies using the iso6ΔC mutant support the possibility that the unique C-t domain of nuclear FMRP isoforms is likely responsible for regulating the stability of the proteins by targeting them to the nuclear proteasome for degradation. However, it remains unclear if this C-t domain is sufficient for proteasomal targeting and whether it confers a novel function to FMRP in terms of regulating the proteasome activity and or the nuclear localisation of its components. In a recent study, it was reported that FMRP deficiency in mice results in an increased activity of the proteasome, which contributes to the neuropathology seen in fragile X syndrome [58]. This excessive proteostasis occurring in the absence of FMRP was considered to be a compensatory mechanism to the overproduction of proteins that generally results from the loss of translation repression mediated by full-length FMRP. The molecular mechanisms underlying the elevation of the proteasome activity in the absence of FMRP, and importantly, the potential role of FMRP, i.e., mediated by its nuclear isoforms in dampening the hyperactivation of the proteasome, remains unknown. Thus, while cytoplasmic full-length FMRP isoforms are involved in the control of protein synthesis, we speculate that future studies investigating the role of nuclear isoforms in regulating the activity of the proteasome may reveal how the cytoplasmic and nuclear FMRP isoforms function in a compensatory pathway to maintain cell homeostasis, and possibly also further extend our understanding the molecular pathways underlying the role of nFMRP in DNA damage signaling by regulating the activity of the nuclear proteasome, whose alteration may contribute to the pathophysiology of FXS.

## 5. Conclusions

This study provides a comprehensive interactome of nuclear FMRP isoforms consisting of proteins involved in DNA, RNA, and protein processes. Our finding that nFMRP associates with factors involved in DNA damage response supports our previous work [18], implicating the protein in such a process. Future investigations should determine how such interactions modulate the function of nFMRP in DNA damage signaling. Similarly, the association of nFMRP with regulators of RNA processing provides us with an opportunity to investigate the potential role of these associations in regulating RNA metabolism, including splicing and processing, towards discovering new functions of FMRP. Finally, while we identified nFMRP as a new target of the proteasome, its specific association with the 19S proteasomal components raises an intriguing possibility of a novel role of nFMRP in regulating proteasome activity or acting with such partners in regulating specific nuclear processes such as DNA damage response. Clearly, this study constitutes an initial step towards defining nFMRP interactions, requiring functional studies to establish the role of nFMRP in maintaining normal cell physiology through specific interactions.

## Figures and Tables

**Figure 1 cells-12-02807-f001:**
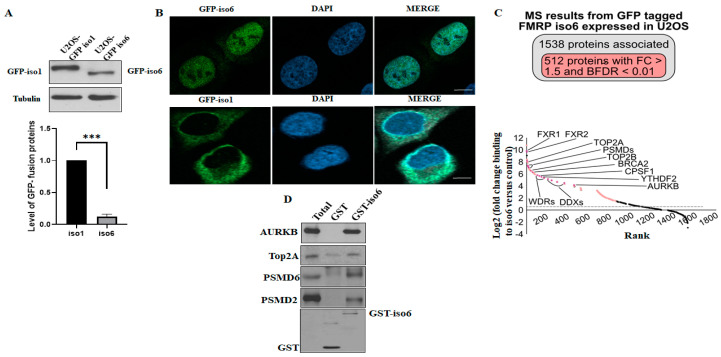
(**A**,**B**) U2OS stably expressing either GFP-iso6 or-iso1 was either lysed and its protein extracts analysed for the expression of GFP-iso6 and -iso1 using anti-GFP antibodies or fixed and processed for immunofluorescence to visualise GFP. Blue staining depicts nuclear DNA. Scale bars (10 μm) are shown. Images were acquired using an LSM 900 confocal microscope. For Western blot quantification, the expression level of GFP-iso6 relative to GFP-iso1 was estimated via densitometry quantification of the film signal using Image Studio™ Lite 4.0.21 software upon standardization against total tubulin. Data are represented as mean ± SD of three independent experiments. A two-tailed Student’s *t* test was used. *** *p* ≤ 0.001. (**C**,**D**) GFP-iso6 interactome. U2OS stably expressing GFP-iso6 or GFP-free U2OS controls were lysed, and their proteins were pulled down in the GFP-trap. Bound proteins from two controls and two experimental GFP-Traps were identified via LC-Ms/Ms. (**C**) Indicated on the top is a total number of proteins pulled down, showing differentially associated proteins in U2OS-GFP iso6 compared to GFP-free U2OS identified associated with GFP-iso6, and those scored with <1% FDR and a fold change higher than 1.5. On the bottom is a graph representing the fold enrichment of selected proteins that bind to iso6. The values are represented on a log2 scale. *p* value ≤ 0.05. (**D**) GST-pull downs. U2OS protein extracts were incubated with GST-iso6 or GST immobilised on glutathione beads. Bound proteins were eluted and analysed via Western blot using antibodies specific to the indicated proteins.

**Figure 2 cells-12-02807-f002:**
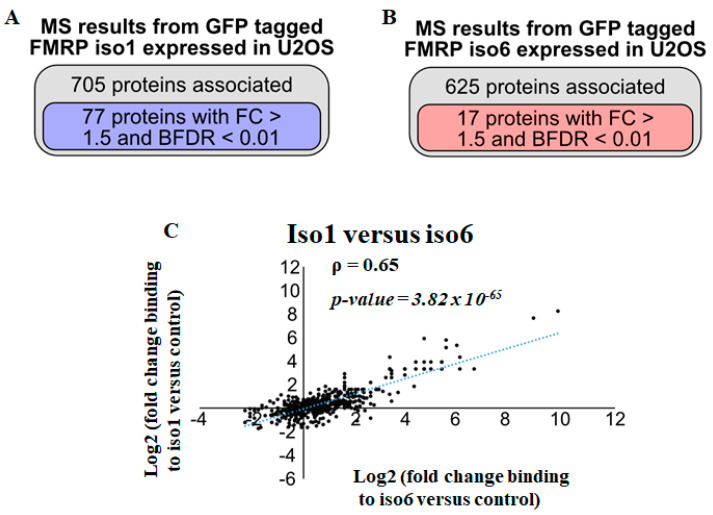
Analysis of GFP-iso6 and GFP-iso1 interactions under stringent conditions. U2OS stably expressing GFP-iso6, -iso1, or GFP-free U2OS controls were lysed, and their proteins pulled down in the GFP-Trap. Bound proteins from 6 controls and 2 experimental experiments were identified via LC-Ms/Ms. (**A**,**B**) The total number of proteins pulled down showing differentially associated proteins in either U2OS-GFP-iso1 (**A**) or -GFP-iso6 is indicated (**B**) compared to GFP-free U2OS, and those scored with <1% FDR and a fold change higher than 1.5. (**C**) Correlation of fold change proteins associated with iso1 and iso6. Pearson correlation coefficients (ρ) and linear regression (blue dotted lines) are indicated.

**Figure 3 cells-12-02807-f003:**
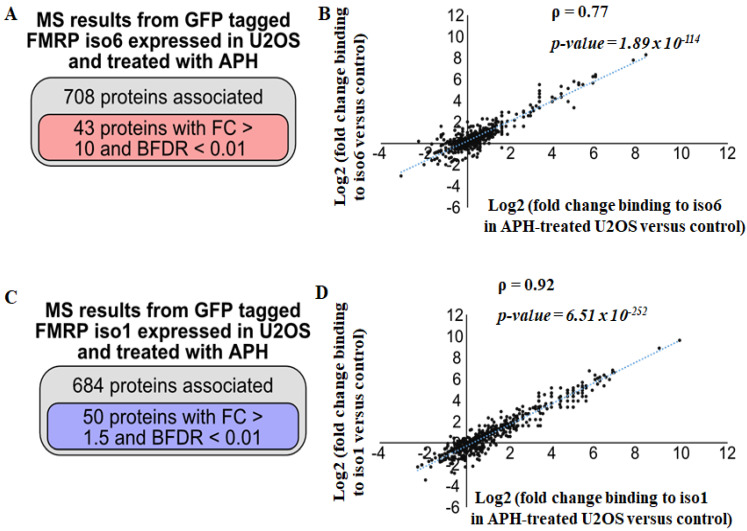
Analysis of GFP-iso6 and GFP-iso1 interactions that occur in APH-treated U2OS under stringent conditions. These experiments were carried out and the data analysed as described in Figure 2, except where U2OS was treated with 0.3 mM APH for 24 h. (**A**) The total number of proteins pulled down in APH-treated U2OS-GFP-iso6 is indicated, showing differentially associated proteins compared to GFP-free U2OS, and those scored with <1% FDR and a fold change higher than 1.5. (**B**) Less correlation in fold change is found between proteins associating with GFP-iso6 in APH-treated U2OS versus mock-treated U2OS. (**C**) The total number of proteins pulled down in APH-treated U2OS-GFP-iso1 is indicated, showing differentially associated proteins compared to GFP-free U2OS, and those scored with <1% FDR and a fold change higher than 1.5. (**D**) A strong correlation in fold change is found between proteins associating with GFP-iso1 in APH-treated U2OS versus mock-treated U2OS.

**Figure 4 cells-12-02807-f004:**
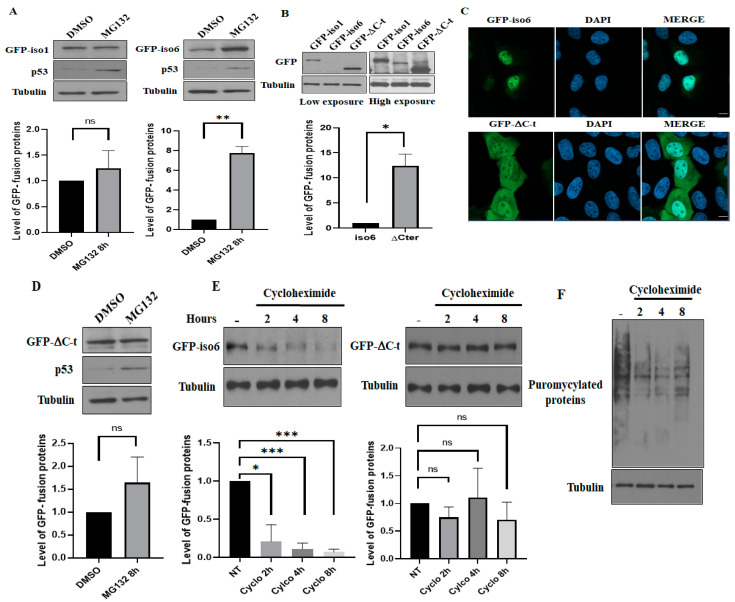
(**A**) Effect of MG132 treatment on the expression of FMRP. U2OS stably expressing either GFP-iso1 or -iso6 was treated with MG132 or DMSO, lysed, and its protein extracts were analysed for the expression of GFP-iso1 (**left** panels) and -iso6 (**right** panels) using anti-GFP antibodies. The expression p53 is used to monitor the effect of MG132, while tubulin serves as a loading control for GFP quantification, as shown on the bottom. For these quantifications, the expression level of GFP-iso1 and -iso6 was estimated via densitometry quantification of the film signal using Image Studio™ Lite 4.0.21 Software and standardized against total tubulin. Data are represented as mean ± SD of three independent experiments. A two-tailed Student’s *t* test was used. ** *p* ≤ 0.01. * *p* ≤ 0.05. ns: not significant. (**B**) The deletion of the C-t domain of iso6 restores the expression of the protein. U2OS stably expressing either GFP-iso1, -iso6, or iso6 (ΔC-t) was lysed, and its protein extracts were analysed for the expression of GFP-iso1 and -iso6 and ΔC-t using anti-GFP (top panels) antibodies. Tubulin (bottom panels) serves as a loading control. The expression level of GFP-ΔC-t relative to -iso6 was estimated via densitometry quantification of the film signal using Image Studio™ Lite Software upon standardization against total tubulin. Data are represented as mean ± SD of three independent experiments. A two-tailed Student’s *t* test was used. * *p* ≤ 0.05. (**C**) The deletion of the C-t domain of iso6 results in a significant cytoplasmic accumulation of the protein. U2OS stably expressing either GFP-iso6 or-ΔC-t were fixed and processed for immunofluorescence to visualise GFP. Blue staining depicts nuclear DNA. Scale bars (5 μm) are shown. Images were acquired by using an LSM 900 confocal microscope. (**D**) U2OS stably expressing GFP-ΔC-t was treated with MG132 or DMSO, lysed, and its protein extracts were analysed for the expression of GFP-ΔC-t using anti-GFP antibodies. The expression p53 is used to monitor the effect of MG132, while tubulin serves as a loading control for GFP quantification, as shown on the bottom. Data are represented as mean ± SD of three independent experiments. A two-tailed Student’s *t* test was used. ns: not significant. (**E**) Treatment with cycloheximide results in a rapid drop of the expression of iso6 but not iso6ΔC-t. U2OS stably expressing either GFP-iso6 or GFP-ΔC-t was treated with cycloheximide for the indicated time before adding puromycin for the last 5 min to monitor ongoing protein synthesis. Cells were then lysed, and their protein extracts were analysed for the expression of GFP-iso6 and ΔC-t using anti-GFP antibodies. Data are represented as mean ± SD of three independent experiments. A two-tailed Student’s *t* test was used. *** *p* ≤ 0.001. * *p* ≤ 0.05. ns: not significant. (**F**) U2OS was treated with cycloheximide for the indicated time. Cells were then lysed, and their protein extracts analysed to probe puromycylated proteins indicating active translation, while tubulin served as a loading control.

**Table 1 cells-12-02807-t001:** Selected proteins associating with iso6 with <1% FDR are grouped in functional groups based on gene ontology annotation.

Functions	Genes	Average Spectral Counts	Fold Change	BFDR
RNA splicing	SRSF2	6	20	0
	SRSF7	6	60	0
	SRSF9	6	60	0
	PRPF3	2	20	0
	PRPF6	6	6	0.01
	PRPF8	44.5	4.45	0
	CPSF1	5	50	0
RNA modification, RNA structure and stability	FXR1/2	90/80	900/800	0
	YTHDF2	3.5	35	0
	DDX50	10	100	0
	DDX52	5.5	55	0
	DDX51	3	30	0
	TRMT1L	3	30	0
Ribosome biogenesis and maturation	WDR3	11.5	115	0
	WDR36	9.5	95	0
	WDR46	3	60	0
	MRTO4	7.5	75	0
	NOP9	2.5	25	0
	NOP16	7	7	0.01
RNA synthesis	POLR2B	8.5	85	0
	POLR1A	6.5	65	0
	POLR1E	3.5	35	0
	CEBPZ	17	170	0
	SUPT16H	7	70	0
	SUPT6H	3	30	0
DNA replication and DNA repair	BRCA2	7	70	0
	TOP2A	25	250	0
	TOP2B	10	100	0
	SMARCA5	8.5	85	0
	SMARCD2	2.5	25	0
	SMARCE1	2	20	0
Chromatin remodeling and chromosome condensation and segregation	H2BC3	33.5	335	0
	H2BC5	34.5	2.88	0.01
	AURKB	2.5	25	0
	RCC1	7.5	75	0
Components of ubiquitin-proteasome system	PSMDs	4.5–29	45–290	0–0.01
	PSMCs	4.5–16.5	5.5–45	0
	UBAP2L	79.5	7.23	0

**Table 2 cells-12-02807-t002:** Ontology of the potential iso6 interactors identified under stringent conditions.

Functions	Genes	Average Spectral Counts	Fold Change	BFDR
RNA binding protein	FXR1	30	300	0
	FXR2	20	200	0
Lipoprotein receptor	LRP1	6	60	0
Ubiquitin-proteasome pathway	PSMD2	6	300	0
	PSMD1	5.5	200	0
	PSMC4	5.5	16.5	0
	PSMD6	3.5	35	0
	PSMD3	3.5	35	0
	PSMD11	3	30	0.01
	PSMD8	2.5	25	0.01
	PSMD13	2.5	25	0.01
	PSMD14	2	20	0.01
Regulation of mitotic spindle orientation	ARHGEF2	5.5	55	0
Mitochondrial metabolism	C1QBP	4	40	0
Transcription	HP1BP3	3.5	35	0
Biosynthesis of phosphatidylinositol	PI4KA	3	30	0.01
Free AA regulation	LPCAT3	2	20	0.01

**Table 3 cells-12-02807-t003:** Fold change analysis of proteins associating with GFP-iso6 in APH-treated U2OS versus mock-treated U2OS.

Genes	Iso6 Fold Change	Iso6 APH Fold Change	Genes	Iso6 Fold Change	Iso6 APH Fold Change
FXR1	300	310	PSMD13	25	NA
FXR2	200	220	CASK	NA	25
PSMD2	60	85	PSMC6	NA	25
LRP1	60	75	PSMD4	NA	25
PSMD1	55	75	NKTR	NA	25
ECT2	NA	60	PSMD14	20	25
ARHGEF	55	50	CPSF1	NA	25
PSMD11	30	55	ADNP	NA	25
KIF22	NA	50	NIP7	NA	25
PSMD6	35	50	MRTO4	NA	25
HMGXB	NA	45	BRD4	NA	25
C1QPB	40	45	LPCAT3	20	20
PSMD3	35	45	RFC2	NA	20
HP1BP3	35	45	PCF11	NA	20
RPL10A	NA	40	NOL7	NA	20
PSMD8	25	35	YLPM1	NA	20
WDR36	NA	35	ITM2B	NA	20
PI4KA	30	NA	TRMT1L	NA	20
APP	NA	30	PLBD2	NA	20
PAK1IP1	NA	30	PPA2	NA	20
TOP2A	NA	30	PSMC4	16.5	18
WDR3	NA	30	AURKB	NA	16.5
			PSMD7	NA	11.25

## Data Availability

Data are contained within the article and supplementary materials.

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
