# Peer review of "The Identification of Nuclear FMRP Isoform Iso6 Partners"

_cells, 2023, doi:10.3390/cells12242807_

Round 1
Reviewer 1 Report
Comments and Suggestions for Authors
The authors studied the binding partners of the nuclear FMRP isoform 6 with special interest in those specifically binding the nuclear isoform and not the cytoplasmic canonical isoform. When carrying out this analysis in U2OS osteosarcoma cell lines using a chimeric N-terminally GFP-fused construct, they identified a few such interactors. Interestingly, among these partners (or partners of partners in larger assemblies) were also found many proteasomal components. When deleting the hypothesized binding domain (C-terminal domain), the association was not detectable. AP-MS yielded higher abundance with proteasome inhibition, and higher degradation with translation inhibition.
In general, I appreciate the research question, this study could be a very important piece. The findings indicate that the analysis is worth more time investment because the authors have important points of discovery. However, I have several major concerns regarding the study design, controls, and I am hereby proposing new experiments. This means, I cannot endorse the paper for publication in its current form due to some major points. Hence, I recommend the authors to put extra work into the study to improve the quality of the paper.
Major points:
· I’m wondering why only iso1,7 and 6,12 were mentioned in the manuscript? It is claimed that iso6 and 12 are the “main ones” in the nucleus, but main ones in what regard? Gene expression or protein abundance in a particular cell type? Actually, I found no experiments in the paper on iso7 and iso12, so the title and abstract should also be adjusted to “nuclear FMRP isoform 6”.
· As I see, a main limitation of the study is that the protein isoforms were expressed in U2OS cells, where FMRP is usually present in very low protein abundance (according to PAXdb). And as FMRP-iso6 is even lower abundance, this choice may not be suitable. This limitation is not properly discussed in the manuscript. In the Discussion chapter, an important HeLa-based FMRP pull-down experiment (Taha et al. 2021) and a rat brain-derived GST pull-down on the nuclear fraction (Kieffer et al. 2022) are mentioned, but as a minimum, more comparative analyses are needed with these datasets – with figures and tables. However, the best would be to also complement the results with a physiologically more relevant cell line on both iso1 and iso6, and compare them with those performed on U2OS cells.
· Another limitation of the study is that the authors failed to demonstrate that GFP-fused FMRP isoforms and wt FMRP isoforms have comparable gene expression and protein abundance. This limitation is again left undiscussed.
· Resolution of the figures is extremely poor. For example, the numbers on Fig. 1D are not readable in column “Average Spectral Counts”, zero, six, eight looks the same.
· The labeling and legend of Fig. 1C and Fig. 2A-B are improper (e.g. “versus U2OS”, “scale bars 10 m”) and confusing, it is not apparent what the difference is between Fig. 1C and Fig. 2B. On one FC is greater or equal than 1.5, on the other FC is greater than 1.5 is chosen as one of the selection criteria.
· How come TOP2A and AURKB are not shown on Fig.2? Weren’t they supposed to be identified as these represent stable validated (Fig. 1E) complexes? Even in the Discussion I haven’t found any mention to these.
· One of the most interesting findings is that the GFP-FMRP-iso6 has specificity to bind proteasomal proteins in U2OS cells, most probably few by direct binding and the rest is pulled down with the rest of the stable complex. The authors proved the binding domain (C-t), however missed the opportunity to [i.] narrow down the binding region; [ii.] determine which proteasomal components are bound directly and indirectly. These could have been done experimentally, or in case of means to do them, even computational approaches could have been tested.
· Has the study performed by Kieffer et al. (2022) also found proteasomal partners for the nuclear FMRP fraction? This answer would be most crucial to the readers, but only other protein categories are discussed.
Minor points:
· In the Suppl. Fig. 1 replace the word “major” by a more informative word. Add abbreviations to the legend.
· Materials and methods 2.1: Mention what the drugs are for, otherwise the interpretability is hampered.
· Fig. 1A: On the bar chart, the error bar displaying the SD is missing from the black bar of iso1.
· In line 356: “quantif”
· All delta are missing from Results 3.4.
Comments on the Quality of English Language
The quality of the language is good in general, only minor editing of the language is required.
Author Response
1-I’m wondering why only iso1,7 and 6,12 were mentioned in the manuscript? It is claimed that iso6 and 12 are the “main ones” in the nucleus, but main ones in what regard? Gene expression or protein abundance in a particular cell type? Actually, I found no experiments in the paper on iso7 and iso12, so the title and abstract should also be adjusted to “nuclear FMRP isoform 6”.
Response: We focused on iso1 and 6 because those are the main studied isoforms. As shown in Supplementary figure 1 and now described in the text, iso1 and iso7 are almost identical with the only difference is the skipping of exon 12 in iso7 (Sittler et al Hum Mol Genet 1996), while the only difference between iso6 and 12 is the skipping of exon 12 in iso12 (Valverde et al Structure 2007). As requested by the reviewer, we have also adjusted the title and abstract to “nuclear FMRP isoform 6”
2- As I see, a main limitation of the study is that the protein isoforms were expressed in U2OS cells, where FMRP is usually present in very low protein abundance (according to PAXdb). And as FMRP-iso6 is even lower abundance, this choice may not be suitable. This limitation is not properly discussed in the manuscript. In the Discussion chapter, an important HeLa-based FMRP pull-down experiment (Taha et al. 2021) and a rat brain-derived GST pull-down on the nuclear fraction (Kieffer et al. 2022) are mentioned, but as a minimum, more comparative analyses are needed with these datasets – with figures and tables. However, the best would be to also complement the results with a physiologically more relevant cell line on both iso1 and iso6, and compare them with those performed on U2OS cells.
Response: U2OS is standard cell line model that is widely used for proteomic and functional studies revealing fundamental molecular and cellular mechanisms, and thus has been used in a number of studies investigating the function of FMRP (Yang et al PNAS 2022; Geng et al JBC 2017; Tang et al. Mol Brain. 2023; Bley et al. Nucleic Acids Res. 2015). Accordingly, using anti-FMRP antibodies (mab1C3) that detect iso1/7 in western blot analyses, we found quantitative expression of these isoforms in U2OS, which seem similarly expressed as housekeeping proteins such as tubulin in U2OS. This is also consistent with FMRP been an abundant protein in various tissues and cell lines (Khandjian et al Hum Mol Genet. 1995; Devys et al Nat Genet. 1993). We have also reported the expression and localization of both iso1/7 and iso6/12 in U2OS, using immunofluorescence with an antibody (C10) that recognizes both type of isoforms (Ledoux et al MBoC 2023). The finding that GFP-iso6 (but not GFP-iso1) is a substrate of the proteasome in U2OS provides an explanation of the observed low abundance of that isoform, though the underlying mechanisms remained to be established. This research question is part of our future studies defining the potential role of nuclear FMRP isoforms in the activity of the proteasome involving interactions with proteasomal proteins using various cell lines including neurons. This point is now discussed in the revised version.
3-Another limitation of the study is that the authors failed to demonstrate that GFP-fused FMRP isoforms and wt FMRP isoforms have comparable gene expression and protein abundance. This limitation is again left undiscussed.
Response: As explained in the text, GFP-iso6 have low abundance likely due to proteasome targeting. At this stage, we cannot determine the abundance of this fusion protein relative to the corresponding endogenous protein, due to the lack of antibodies that detect quantitatively endogenous nuclear FMRP isoforms. We are investing efforts to develop such antibodies that will certainly be highly useful for both expression, localisation and functional studies of nuclear FMRP isoforms by the scientific community.
4-Resolution of the figures is extremely poor. For example, the numbers on Fig. 1D are not readable in column “Average Spectral Counts”, zero, six, eight looks the same.
Response: We have corrected this in figures 1D, 2C, and 3B.
5-The labeling and legend of Fig. 1C and Fig. 2A-B are improper (e.g. “versus U2OS”, “scale bars 10 m”) and confusing, it is not apparent what the difference is between Fig. 1C and Fig. 2B. On one FC is greater or equal than 1.5, on the other FC is greater than 1.5 is chosen as one of the selection criteria.
Response: We have corrected this as follows:
Fig.1C: Ms results from GFP tagged FMRP iso6 expressed in U2OS;
Fig.2A: Ms results from GFP tagged FMRP iso1 expressed in U2OS;
Fig.2B: Ms results from GFP tagged FMRP iso6 expressed in U2OS;
Fig.3A: Ms results from GFP tagged FMRP iso6 expressed in U2OS expressed in U2OS and treated with APH;
Fig.3D: Ms results from GFP tagged FMRP iso1 expressed in U2OS and treated with APH;
We have corrected FC to greater to 1.5 in both experiments.
6-How come TOP2A and AURKB are not shown on Fig.2? Weren’t they supposed to be identified as these represent stable validated (Fig. 1E) complexes? Even in the Discussion I haven’t found any mention to these.
Response: Data shown in figure 2 have been obtained using more stringent conditions as compared to those presented in figure 1. Under these stringent conditions, we did not detected Top2A and AURKB in either iso6- or iso1-interactome. These results suggest that binding of Top2A and AURKB with FMRP may not be sufficiently stable under normal growth conditions to be detected under stringent conditions. However, we significantly detected interaction of iso6 (but not iso1) with both Top2A and AURKB under similar stringent conditions in U2OS treated with APH, indicating that replicative stress induced during APH treatment stabilises such interactions. We have clarified this in the discussion section of our our revised version.
7-One of the most interesting findings is that the GFP-FMRP-iso6 has specificity to bind proteasomal proteins in U2OS cells, most probably few by direct binding and the rest is pulled down with the rest of the stable complex. The authors proved the binding domain (C-t), however missed the opportunity to [i.] narrow down the binding region; [ii.] determine which proteasomal components are bound directly and indirectly. These could have been done experimentally, or in case of means to do them, even computational approaches could have been tested.
Response: These are important questions to address, which deserves a complete and independent future study. In such follow-up study, we will define the motifs and amino acids required for the association of FMRP iso6 with PSMDs and PSMCs, determine how these association are regulated i.e. through posttranslational modifications, and provide a comprehensive catalogue of direct and indirect interaction between FMRP and PSMD/PSMC using reciprocal immunoprecipitation/GST pull down, and immunoprecipitation/GST pull downs in cell lines where each interacting PSMD/PSMC is depleted.
8-Has the study performed by Kieffer et al. (2022) also found proteasomal partners for the nuclear FMRP fraction? This answer would be most crucial to the readers, but only other protein categories are discussed.
Response: Our search in Kieffer et al. (2022) data did not revealed any listed proteasomal partner for FMRP in the nuclear rat brain fraction. It should be noted that the study of Kieffer et al. (2022) used the cytoplasmic FMRP isoforms as bait to identify nuclear rat partners present in rat brain nuclear extracts since it was assumed that a fraction (2%) of such isoforms can travel to the nucleus. The lack of PSMD/PSMC as FMRP partners in Kieffer et al. study is consistent with our finding that PSMD/PSMC are specific partners of the nuclear FMRP isoforms, though it may also suggest that such interactions may not occur in murine. Studies investigating the interaction of FMRP nuclear isoforms with proteosomal proteins in murine brain should help resolve this issue.
10-In the Suppl. Fig. 1 replace the word “major” by a more informative word. Add abbreviations to the legend.
Response: We have replaced the word “major” by “best characterised”
11-Materials and methods 2.1: Mention what the drugs are for, otherwise the interpretability is hampered.
Response: By drugs, we refer to puromycine. This is now corrected.
12- Fig. 1A: On the bar chart, the error bar displaying the SD is missing from the black bar of iso1.
Response : Iso1 was used as a reference control with a value of 1 of each replicate.
13-In line 356: “quantif”
Response: We corrected this typo.
14-All delta are missing from Results 3.4.
We have corrected these typos.
Reviewer 2 Report
Comments and Suggestions for Authors
The paper "Identification of the Nuclear FMRP Isoforms Partners" presents an interesting exploration into the nuclear isoforms of the Fragile X Mental Retardation Protein (FMRP) and their interacting partners. The research addresses an important area of neurobiology and contributes valuable insights into the molecular mechanisms underlying Fragile X Syndrome.
Major comments:
1.While the data are well-presented, a deeper discussion of the functional implications of the identified partners would enhance the paper's impact. How do these interactions relate to the pathophysiology of Fragile X Syndrome, and what are the broader implications? For example, the author lists a lot of interacted proteins in the 3.1 and 3.2 sections but does not describe the details of the functional of those interactomes.
2. Only two biological replicates (2.9 section) for each group are not enough for the conclusion. In general, three biological replicates are needed. The lack of sufficient biological replicates reduces the confidence in the data.
3.The authors are requested to provide results verifying the interacting proteins to further support the mass spectrometry results.
Minor comments:
1.The figures are not sharp enough and most pictures are difficult to see clearly. Please provide high-resolution images.
Author Response
We thank this reviewer for their positive and constructive comments to help improving our manuscript.
Our responses (in bold font style) are as follow:
- While the data are well-presented, a deeper discussion of the functional implications of the identified partners would enhance the paper's impact. How do these interactions relate to the pathophysiology of Fragile X Syndrome, and what are the broader implications? For example, the author lists a lot of interacted proteins in the 3.1 and 3.2 sections but does not describe the details of the functional of those interactomes.
Response : At this stage, it is difficult to deeply discuss functional implications of the identified partners in the pathophysiology of the syndrome, without providing functional validations. As now described in the revised version, our future plan is to design functional studies assessing the role of FMRP interactions with the identified partners in neurone proliferation and differentiation using suitable cell lines models such as human pluripotent embryonic stem cells, which are valuable tools to study neuronal processes.
- Only two biological replicates (2.9 section) for each group are not enough for the conclusion. In general, three biological replicates are needed. The lack of sufficient biological replicates reduces the confidence in the data.
Response: We have followed protocols described in previous literatures (Agbo et al J Proteome Res. 2023 PMID: 36484504; Loehr er al Front Mol Biosci. 2022 PMID: 35402505; Go et al Nature. 2021 PMID: 34079125; Lambert et al Mol Cell. 2019 PMID: 30554943; Savitsky et al Cell Rep. 2016 PMID: 27926874; Feng et al EMBO J. 2016 PMID: 26620551; Lambert et al . J Proteomics. 2015 PMID: 25281560; Lambert et al J Proteomics. 2014 PMID: 24412199), describing proteomic data obtained with two biological replicates and analysed by SAINTexpress.
3.The authors are requested to provide results verifying the interacting proteins to further support the mass spectrometry results.
Response : As validation, we have produced GST-pull down results showing interactions of GST with selected partners (Fig. 1D). We have now included new data showing that interaction of GFP-iso6 with selected proteins in GFP-trap and by GST pull down showing that GST-iso6 interactions with selected partners are likely direct, not mediated by RNA (Supplementary figure. 2A-B).
Minor comments:
1.The figures are not sharp enough and most pictures are difficult to see clearly. Please provide high-resolution images.
Response: We have now produced our figures with higher resolution.
Reviewer 3 Report
Comments and Suggestions for Authors
The manuscript authored by Ledoux et al. comprehensively identifies the interacting proteins of nuclear FMRP isoform, FMRP iso6, by affinity purification coupled with mass spectrometry and provides insight into the nuclear role of FMRP in human osteosarcoma (U2OS) cells. Interestingly, by comparing nuclear and cytoplasmic FMRP interactomes, authors identified that the nuclear FMRP iso6 isoform specifically interacts with many proteasome proteins, such as PSMD and PSMC components. Furthermore, the authors define that the C-terminal domain of FMRP iso6 is responsible for being a target of proteasome-mediated protein degradation, while the physiological meaning of this rapid protein degradation of FMRP iso6 remains unclear.
While most of the figures need to be replaced with a clear resolution, their experimental observations are very impressive and offer novel information on nuclear FMRP function.
Addressing the following specific concerns will significantly enhance the rigor and clarity of the manuscript.
1. Figures. Most figures suffer from low resolution, making it difficult to discern details, and the fonts used in figures are hard to recognize. Please replace these figures with higher-quality images to improve clarity and readability.
2. Title: The current title is vague and lacks novelty. Consider revising it to provide a more specific description of the findings to make the paper's contribution clearer.
3. Abstract, lines 24-25, and Introduction. Previous literature, particularly the work of Bardoni et al. (Hum Mol Genet 1999), identified nuclear FMRP interacting protein (NUFIP). Properly integrate this information into the manuscript for context and completeness.
4. Page 3, lines 110-116, and Page 9 lines 346-348. In experiments validating direct interactions between FMRP iso6 and its partners, it is advisable to perform assays under RNase-treated conditions, such as using turbonuclease as described in the Affinity Purification and MS Experiments section. Ensure consistency in experimental conditions and clarify the use of turbonuclease in GST-pull down assays.
5. Figure 1A. Include a comparison between GFP-FMRP and endogenous FMRP levels to demonstrate that GFP-FMRP expression is not excessively elevated, ensuring the physiological relevance of the expression levels.
6. Figure 1B and page 5, lines 225-226. Address the disparity between the observed nucleoplasmic localization of GFP-FMRP iso6 in Figure 1B and the previous finding by Dury et al. (Dury et al., Plos Genet 2013) that FMRP iso6 predominantly localizes to Cajal bodies. Clarify if GFP-FMRP iso6 was observed in Cajal bodies.
7. Page 12, lines 480. Verify the absence of BRD2 in the provided list (Supplementary Table 1 and Fig. 1D-E) to ensure accuracy in the data presented.
8. Figure 4F and lines 453-454. Reevaluate Figure 4F, as the 8-hour cycloheximide treatment appears to indicate translation recovery, causing confusion. Consider revising or replacing this figure to present the data more clearly.
Author Response
We thank this reviewer for their positive and constructive comments to help improving our manuscript.
Our responses (in bold font style) are as follow:
- Figures. Most figures suffer from low resolution, making it difficult to discern details, and the fonts used in figures are hard to recognize. Please replace these figures with higher-quality images to improve clarity and readability.
Response: We have now produced our figures with higher resolution.
- Title: The current title is vague and lacks novelty. Consider revising it to provide a more specific description of the findings to make the paper's contribution clearer.
Responses: we have replaced our title by: Identification of the Nuclear FMRP Isoform iso6 Partners
- Abstract, lines 24-25, and Introduction. Previous literature, particularly the work of Bardoni et al. (Hum Mol Genet 1999), identified nuclear FMRP interacting protein (NUFIP). Properly integrate this information into the manuscript for context and completeness.
Response: We have included this finding in the introduction section.
- Page 3, lines 110-116, and Page 9 lines 346-348. In experiments validating direct interactions between FMRP iso6 and its partners, it is advisable to perform assays under RNase-treated conditions, such as using turbonuclease as described in the Affinity Purification and MS Experiments section. Ensure consistency in experimental conditions and clarify the use of turbonuclease in GST-pull down assays.
Response: We have included new RNase data (Supplementary data 2B) excluding the possibility that interaction of iso6 with selected partners is mediated by RNA.
- Figure 1A. Include a comparison between GFP-FMRP and endogenous FMRP levels to demonstrate that GFP-FMRP expression is not excessively elevated, ensuring the physiological relevance of the expression levels.
Response: We and others have previously shown that overexpression of FMRP induces FMRP-containing RNA granules such as stress granules that may affect the activity and localisation of the protein (Mazroui et al HMG 2002; Dury et al Plos Gnetcios 2-13). To avoid the induction of this phenomenon due to excessive expression of GFP proteins, we have selected U2OS clones that do not show such formation of FMRP granules, indicating moderate expression. To our best knowledge, there is no antibody available that recognizes specifically nFMRP, while the IgY#C10, is the only antibody known to detect all known FMRP isoforms, which we and other have validated as a unique and suitable antibody to study nFMRP (Dury et al., 2013), in particular using immunofluorescence experiments. However, using this unique antibody do not allow quantitative assessment of the expression level of nFMRP in western blot of total extracts prepared from various human somatic cells, presumably due to their low-expression, as compared to cFMRP.
- Figure 1B and page 5, lines 225-226. Address the disparity between the observed nucleoplasmic localization of GFP-FMRP iso6 in Figure 1B and the previous finding by Dury et al. (Dury et al., Plos Genet 2013) that FMRP iso6 predominantly localizes to Cajal bodies. Clarify if GFP-FMRP iso6 was observed in Cajal bodies.
Response: Dury et al study reported the localisation of GFP-iso6 in cajal bodies formed in HeLa and mouse embryonic fibroblasts. In their studies, Dury et al observed that localisation of the protein upon acute (6 hours) transient expression to avoid saturation of the nuclei by the protein. Whether similar localisation of the protein occurs upon prolonged transient expression was not addressed. In our study, we used GFP-iso6 that is stably expressed in U2OS and thus its expression is maintained for days before localisation studies. Although we occasionally observed GFP-iso6 localised in cajal bodies, this association with cajal bodies was not consistently evident probably reflecting a transient association of the protein with those foci. The possibility that the expression level of iso6 affect its association with cajal bodies is a possibility that deserves further studies. This is now clarified in the text. The new figure (included in the attached file of this response letter) shows the localisation of GFP-iso6 that we transiently expressed in U2OS and validated its localisation in nuclear foci resembling cajal bodies, as was described in Dury et al 2013 study. We can include this data in our revised manuscript if upon request.
- Page 12, lines 480. Verify the absence of BRD2 in the provided list (Supplementary Table 1 and Fig. 1D-E) to ensure accuracy in the data presented.
Response: We apologise for the writing mistake. Our data revealed BRD4 (and not BRD2) as a partner of nFMRP. This is now corrected in our revised version.
- Figure 4F and lines 453-454. Reevaluate Figure 4F, as the 8-hour cycloheximide treatment appears to indicate translation recovery, causing confusion. Consider revising or replacing this figure to present the data more clearly.
Response; We have now produced new blots (Fig. 4F) that clearly show inhibition of translation in cycloheximide-treated U2OS.

Round 2
Reviewer 1 Report
Comments and Suggestions for Authors
Earlier major points:
1-I’m wondering why only iso1,7 and 6,12 were mentioned in the manuscript? It is claimed that iso6 and 12 are the “main ones” in the nucleus, but main ones in what regard? Gene expression or protein abundance in a particular cell type? Actually, I found no experiments in the paper on iso7 and iso12, so the title and abstract should also be adjusted to “nuclear FMRP isoform 6”.
Response: We focused on iso1 and 6 because those are the main studied isoforms. As shown in Supplementary figure 1 and now described in the text, iso1 and iso7 are almost identical with the only difference is the skipping of exon 12 in iso7 (Sittler et al Hum Mol Genet 1996), while the only difference between iso6 and 12 is the skipping of exon 12 in iso12 (Valverde et al Structure 2007). As requested by the reviewer, we have also adjusted the title and abstract to “nuclear FMRP isoform 6”
ïƒ OK, I see, so "main ones" meant that these are the best studied ones.
2- As I see, a main limitation of the study is that the protein isoforms were expressed in U2OS cells, where FMRP is usually present in very low protein abundance (according to PAXdb). And as FMRP-iso6 is even lower abundance, this choice may not be suitable. This limitation is not properly discussed in the manuscript. In the Discussion chapter, an important HeLa-based FMRP pull-down experiment (Taha et al. 2021) and a rat brain-derived GST pull-down on the nuclear fraction (Kieffer et al. 2022) are mentioned, but as a minimum, more comparative analyses are needed with these datasets – with figures and tables. However, the best would be to also complement the results with a physiologically more relevant cell line on both iso1 and iso6, and compare them with those performed on U2OS cells.
Response: U2OS is standard cell line model that is widely used for proteomic and functional studies revealing fundamental molecular and cellular mechanisms, and thus has been used in a number of studies investigating the function of FMRP (Yang et al PNAS 2022; Geng et al JBC 2017; Tang et al. Mol Brain. 2023; Bley et al. Nucleic Acids Res. 2015). Accordingly, using anti-FMRP antibodies (mab1C3) that detect iso1/7 in western blot analyses, we found quantitative expression of these isoforms in U2OS, which seem similarly expressed as housekeeping proteins such as tubulin in U2OS. This is also consistent with FMRP been an abundant protein in various tissues and cell lines (Khandjian et al Hum Mol Genet. 1995; Devys et al Nat Genet. 1993). We have also reported the expression and localization of both iso1/7 and iso6/12 in U2OS, using immunofluorescence with an antibody (C10) that recognizes both type of isoforms (Ledoux et al MBoC 2023). The finding that GFP-iso6 (but not GFP-iso1) is a substrate of the proteasome in U2OS provides an explanation of the observed low abundance of that isoform, though the underlying mechanisms remained to be established. This research question is part of our future studies defining the potential role of nuclear FMRP isoforms in the activity of the proteasome involving interactions with proteasomal proteins using various cell lines including neurons. This point is now discussed in the revised version.
ïƒ I know U2OS is a standard cell line model, and I appreciate that FMRP was possible to detect in U2OS cells. However, FMRP levels vary a lot across cell lines, and as U2OS is a cancer cell line with various protein levels perturbed, the interactome detected in U2OS may substantially differ from that in another cell line. Therefore, I think the ways to resolve this critical point are (1) either to emphasize in the Abstract and Introduction that the interactome reported is from U2OS cell, (2) or to prove that the interactome is essentially the same in a non-cancerous cell line.
3-Another limitation of the study is that the authors failed to demonstrate that GFP-fused FMRP isoforms and wt FMRP isoforms have comparable gene expression and protein abundance. This limitation is again left undiscussed.
Response: As explained in the text, GFP-iso6 have low abundance likely due to proteasome targeting. At this stage, we cannot determine the abundance of this fusion protein relative to the corresponding endogenous protein, due to the lack of antibodies that detect quantitatively endogenous nuclear FMRP isoforms. We are investing efforts to develop such antibodies that will certainly be highly useful for both expression, localisation and functional studies of nuclear FMRP isoforms by the scientific community.
ïƒ Other than through antibodies, there are also other techniques to infer comparable levels of nuclear proteins, e.g. via quantitative proteomics on isolated nuclei. However, I accept if the authors include this limitation to the article text.
4-Resolution of the figures is extremely poor. For example, the numbers on Fig. 1D are not readable in column “Average Spectral Counts”, zero, six, eight looks the same.
Response: We have corrected this in figures 1D, 2C, and 3B.
ïƒ Great
5-The labeling and legend of Fig. 1C and Fig. 2A-B are improper (e.g. “versus U2OS”, “scale bars 10 m”) and confusing, it is not apparent what the difference is between Fig. 1C and Fig. 2B. On one FC is greater or equal than 1.5, on the other FC is greater than 1.5 is chosen as one of the selection criteria.
Response: We have corrected this as follows:
Fig.1C: Ms results from GFP tagged FMRP iso6 expressed in U2OS;
Fig.2A: Ms results from GFP tagged FMRP iso1 expressed in U2OS;
Fig.2B: Ms results from GFP tagged FMRP iso6 expressed in U2OS;
Fig.3A: Ms results from GFP tagged FMRP iso6 expressed in U2OS expressed in U2OS and treated with APH;
Fig.3D: Ms results from GFP tagged FMRP iso1 expressed in U2OS and treated with APH;
We have corrected FC to greater to 1.5 in both experiments.
ïƒ If Fig. 1C and 2B are the same, it is still not apparent, why they have a total of 1538 vs 625 proteins associated, respectively.
6-How come TOP2A and AURKB are not shown on Fig.2? Weren’t they supposed to be identified as these represent stable validated (Fig. 1E) complexes? Even in the Discussion I haven’t found any mention to these.
Response: Data shown in figure 2 have been obtained using more stringent conditions as compared to those presented in figure 1. Under these stringent conditions, we did not detected Top2A and AURKB in either iso6- or iso1-interactome. These results suggest that binding of Top2A and AURKB with FMRP may not be sufficiently stable under normal growth conditions to be detected under stringent conditions. However, we significantly detected interaction of iso6 (but not iso1) with both Top2A and AURKB under similar stringent conditions in U2OS treated with APH, indicating that replicative stress induced during APH treatment stabilises such interactions. We have clarified this in the discussion section of our our revised version.
ïƒ The term “under stringent conditions” is a recurring expression throughout the manuscript, however, it is poorly explained.
7-One of the most interesting findings is that the GFP-FMRP-iso6 has specificity to bind proteasomal proteins in U2OS cells, most probably few by direct binding and the rest is pulled down with the rest of the stable complex. The authors proved the binding domain (C-t), however missed the opportunity to [i.] narrow down the binding region; [ii.] determine which proteasomal components are bound directly and indirectly. These could have been done experimentally, or in case of means to do them, even computational approaches could have been tested.
Response: These are important questions to address, which deserves a complete and independent future study. In such follow-up study, we will define the motifs and amino acids required for the association of FMRP iso6 with PSMDs and PSMCs, determine how these association are regulated i.e. through posttranslational modifications, and provide a comprehensive catalogue of direct and indirect interaction between FMRP and PSMD/PSMC using reciprocal immunoprecipitation/GST pull down, and immunoprecipitation/GST pull downs in cell lines where each interacting PSMD/PSMC is depleted.
ïƒ Accepted.
8-Has the study performed by Kieffer et al. (2022) also found proteasomal partners for the nuclear FMRP fraction? This answer would be most crucial to the readers, but only other protein categories are discussed.
Response: Our search in Kieffer et al. (2022) data did not revealed any listed proteasomal partner for FMRP in the nuclear rat brain fraction. It should be noted that the study of Kieffer et al. (2022) used the cytoplasmic FMRP isoforms as bait to identify nuclear rat partners present in rat brain nuclear extracts since it was assumed that a fraction (2%) of such isoforms can travel to the nucleus. The lack of PSMD/PSMC as FMRP partners in Kieffer et al. study is consistent with our finding that PSMD/PSMC are specific partners of the nuclear FMRP isoforms, though it may also suggest that such interactions may not occur in murine. Studies investigating the interaction of FMRP nuclear isoforms with proteosomal proteins in murine brain should help resolve this issue.
ïƒ OK
New minor points and spelling mistakes:
- Figures should be inserted after paragraphs where they are first mentioned.
- Lines 283-285: This is more like discussion than results.
- Lines 248, 254, 512: "cajal bodies" --> "Cajal bodies"
- Line 302: "puled-down" --> "pulled down"
- Line 325 (Fig. 3D): "APH reated" --> "APH-treated"
- Line 383: extra comma to be removed
- Line 502: puled --> pulled
- Line 517: Omic --> omics
- Lines 525, 530, 536: neurone --> neuron
- Line 683: ".." --> "."
- Line 686: isoformes --> isoforms
- Line 693: target --> interaction partner (to avoid confusion about being a substrate for degradation)
Comments on the Quality of English Language
Mainly just minor edits with a very throrough proofreading are required.
Author Response
We thank the reviewer for recommendations helping improving the manuscript.
1- I know U2OS is a standard cell line model, and I appreciate that FMRP was possible to detect in U2OS cells. However, FMRP levels vary a lot across cell lines, and as U2OS is a cancer cell line with various protein levels perturbed, the interactome detected in U2OS may substantially differ from that in another cell line. Therefore, I think the ways to resolve this critical point are (1) either to emphasize in the Abstract and Introduction that the interactome reported is from U2OS cell, (2) or to prove that the interactome is essentially the same in a non-cancerous cell line.
Response: We have now indicated in both the abstract and the introduction sections U2OS as the cell line used in our study.
2-Other than through antibodies, there are also other techniques to infer comparable levels of nuclear proteins, e.g. via quantitative proteomics on isolated nuclei. However, I accept if the authors include this limitation to the article text.
Response: We totally agree with this comment. We have now included this limitation in the text.
In the results section, we added: " Because of the lack of suitable antibodies that can detect quantitatively and specifically endogenous iso6 in western blots using total cell extracts, we could not provide comparison between GFP-FMRP iso6 and endogenous FMRP iso6 levels to demonstrate that GFP-FMRP expression is within a physiological range. However, our immunofluorescence experiments (images are provided to show differential localisation and not differential expression between iso6 and iso1) confirmed the nuclear and cytoplasmic localisation of GFP-iso6 and iso1 (Fig. 1B), respectively".
In the discussion section, we added: "We are aware however that increasing the level of iso6 in U2OS by expressing GFP-iso6 may lead to an overestimation of interactions. Comparative studies assessing the expression of iso6 in U2OS-expressing GFP-iso6 relative to U2OS by quantitative LC-MSMS on isolated nuclei or by translatomic studies quantifying the association of iso 6 RNA with translating polyribosomes, may help address this issue, though our immunofluorescence experiments (Fig. 1B) suggest moderate expression of GFP-iso6 in our U2OS clones we selected for interaction studies".
3-The term “under stringent conditions” is a recurring expression throughout the manuscript, however, it is poorly explained
Response: We have now explained “stringent conditions” in the new version of this manuscript as follows: "In experiment 2 (Fig. 2) assessing differential interactions between iso6 and iso1, we employed half the number of cells used in the experiments for GFP-iso6 pull down described in figure 1C, while including six independent sets of GFP-free U2OS controls as compared to two sets used in the GFP-trap reported in figure 1. This experimental design remained effective at eliminating weak and potentially transient interactions in these stringent conditions". The change to our experimental design was necessary as we lacked key cell culture reagents to perform at our initial scale the experiments when they were performed.
Additional corrections:
We replaced cajal bodies with Cajal bodies.
We replaced "puled-down" with "pulled down"
We replaced "APH reated" with "APH-treated" in figure 3D.
We replaced Omic with omics
We replaced neurone with neuron
We replaced isoformes with isoforms
Reviewer 3 Report
Comments and Suggestions for Authors
This significantly improved manuscript is now a delight to read. The authors have done an excellent job of addressing my concerns and have incorporated relevant literature to highlight their findings. I only have a few suggestions for textual modifications, as indicated below:
Page 13, lines 521-527: I suggest changing the term "FXR" to "FXS" as defined in the Introduction.
Page 14, line 563, and Page 16, line 683: I recommend changing the term "FX syndrome" to "FXS" for consistency in terminology throughout the manuscript.
Author Response
We thank the reviewer for recommendations helping improving the manuscript.
1-Page 13, lines 521-527: I suggest changing the term "FXR" to "FXS" as defined in the Introduction.
Response: We agree with this suggestion and have accordingly changed the term FXR to FXS.
2-I recommend changing the term "FX syndrome" to "FXS" for consistency in terminology throughout the manuscript.
Response: We have changed the term "FX syndrome" to "FXS"